# Determining composition of micron-scale protein deposits in neurodegenerative disease by spatially targeted optical microproteomics

Kevin C Hadley[1†], Rishi Rakhit[2†], Hongbo Guo[3], Yulong Sun[1], James EN Jonkman[4], Joanne McLaurin[5], Lili-Naz Hazrati[6], Andrew Emili[3], Avijit Chakrabartty[7*]

[1]Department of Medical Biophysics, Princess Margaret Cancer Centre, University of Toronto, Toronto, Canada; [2]Department of Chemical and Systems Biology, Stanford University, Stanford, United States; [3]The Banting and Best Department of Medical Research, Terrence Donnelly Centre for Cellular & Biomolecular Research, Department of Molecular Genetics, University of Toronto, Toronto, Canada; [4]Advanced Optical Microscopy Facility, Princess Margaret Cancer Centre, University Health Network, Toronto, Canada; [5]Department of Laboratory Medicine and Pathobiology, University of Toronto, Toronto, Canada; [6]Tanz Centre for Research in Neurodegenerative Diseases, Department of Laboratory Medicine and Pathobiology, University of Toronto, Toronto, Canada; [7]Departments of Biochemistry and Medical Biophysics, Princess Margaret Cancer Centre, University of Toronto, Toronto, Canada

**Abstract** Spatially targeted optical microproteomics (STOMP) is a novel proteomics technique for interrogating micron-scale regions of interest (ROIs) in mammalian tissue, with no requirement for genetic manipulation. Methanol or formalin-fixed specimens are stained with fluorescent dyes or antibodies to visualize ROIs, then soaked in solutions containing the photo-tag: 4-benzoylbenzyl-glycyl-hexahistidine. Confocal imaging along with two photon excitation are used to covalently couple photo-tags to all proteins within each ROI, to a resolution of 0.67 µm in the xy-plane and 1.48 µm axially. After tissue solubilization, photo-tagged proteins are isolated and identified by mass spectrometry. As a test case, we examined amyloid plaques in an Alzheimer's disease (AD) mouse model and a post-mortem AD case, confirming known plaque constituents and discovering new ones. STOMP can be applied to various biological samples including cell lines, primary cell cultures, ex vivo specimens, biopsy samples, and fixed post-mortem tissue.

**\*For correspondence:**
chakrab@uhnresearch.ca

[†]These authors contributed equally to this work

**Competing interests:** The authors declare that no competing interests exist.

## Introduction

Pathological protein deposits have a long history as hallmarks of neurodegenerative disease. Early methods used to identify these deposits include the silver stain introduced by *Golgi (1873)*, and Virchow's iodine–sulfuric acid stain for starch (*Virchow, 1854*) that led to the coining of the term amyloid (*Virchow, 1855*). More specialized stains and labels have emerged which have begun to probe the structure and composition of pathological protein deposits. The stains Congo red and thioflavin S (ThS) were discovered later and remain in use (reviewed by Westermark (*Sipe and Westermark, 2005*) and Tanskanen (*Tanskanen, 2013*)), specifically identifying deposits containing a particular structural motif: amyloid β-pleated sheets. Beyond mere detection lies comprehensive identification of the protein components of these deposits; biochemical analyses of the deposits have

**eLife digest** Neurodegenerative diseases such as Alzheimer's disease affect millions of people worldwide. In many of these diseases, toxic proteins accumulate in the brain and build up as small 'plaques' in the gaps, or synapses, that cells called neurons communicate across. Eventually, the plaques prevent the neurons signaling to each other correctly, leading to problems such as memory loss.

Identifying the proteins present in plaques is technically challenging, partly because the plaques are very small. Hadley, Rakhit et al. have now developed a new method called spatially targeted optical microproteomics (or STOMP) that can collect proteins from small areas of cells. In this method, plaques are identified under a light microscope, and their contents are attached to a molecule called a photo-affinity tag using lasers. The photo-tagged proteins are then pulled out using beads that specifically bind to the photo-affinity tag. The proteins can then be identified using a well-established method called mass spectrometry.

Hadley, Rakhit et al. used STOMP to analyze plaques present in the brains of mice that develop similar symptoms to those seen in humans with Alzheimer's disease. This revealed that these plaques contain more than 50 different proteins, some of which had not previously been found in plaques. In particular, several proteins from the 'presynaptic' neuron that sends signals across the synapse were found in the plaques. However, no proteins from the receiving ('postsynaptic') neuron on the other side of the synapse were present in the plaque.

Fixed human brain tissue is more difficult to analyze than mouse samples because it is modified for storage. In spite of these issues, Hadley, Rakhit et al. successfully also used STOMP to identify the proteins in human plaques. STOMP can be used to identify the proteins present in any area of a cell and thus has the potential to be widely used by scientists, not just those studying plaques.

produced transformative results in the field of neurodegeneration research. Two classic examples include the discovery of the prion protein and formulation of the protein-only hypothesis of prion disease (*Bolton et al., 1982*), and the discovery of the Alzheimer amyloid peptide (Aβ) (*Glenner and Wong, 1984*) and formulation of the amyloid cascade hypothesis of Alzheimer's disease (AD, reviewed by Musiek and Holtzman (*Musiek and Holtzman, 2015*)). More recently, there was the discovery that the RNA/DNA binding protein TDP-43 (transactive response DNA binding protein 43 kDa) is a significant component of ubiquitin-positive intraneuronal inclusions in certain cases of frontotemporal dementia (FTD) and amyotrophic lateral sclerosis (ALS) (*Neumann et al., 2006*). This discovery helped establish these diseases as spectrum disorders (*Mackenzie, 2007*), it led to the identification of TDP-43 mutations that are causative for ALS and FTD (*Sreedharan et al., 2008*) and established a role for RNA metabolism in ALS/FTD pathogenesis (*Lagier-Tourenne and Cleveland, 2009*; *Mackenzie et al., 2010*).

Identification of the protein components of pathological deposits has traditionally involved the partial biochemical purification of detergent-insoluble proteins present in a tissue specimen followed by protein sequencing. Immunohistochemical methods are then used to confirm that the identified proteins are bona fide constituents of the pathological deposits. These methods require large amounts of sometimes-scarce pathological tissue and identification of novel protein components can require increasingly elaborate, time-consuming, and costly protocols. For example, the discovery of TDP-43 in ubiquitin-positive intraneuronal inclusions required the generation of ~1000 monoclonal antibodies, which were used to screen thousands of tissue sections by immunohistochemistry, to obtain a single monoclonal antibody that specifically labeled ubiquitin-positive intraneuronal inclusions (*Neumann et al., 2006*). That antibody was used for the proteomic identification of TDP-43 from the detergent-insoluble fraction of pathological tissue homogenates, and confirmatory staining with commercial TDP-43 antibodies was required for validation. In addition to being costly and labor intensive, another major limitation of biochemical purification of protein deposits is that soluble protein components associated with the core aggregates are likely to be lost during fractionation. Identifying deposit-associated soluble proteins could further our understanding of disease mechanisms.

A different strategy that can preserve some of the deposit-associated soluble proteins is laser capture microdissection (LCM), in which protein deposits are lifted intact out of sections of the fixed

specimen; their protein compositions are then determined by mass spectrometry (*Gozal et al., 2006*). The resolution of LCM is ~10 μm in the horizontal plane and captures the entire thickness of the tissue section along the vertical axis. Thus, for features smaller than ~10 μm—such as inclusion bodies, small amyloid plaques, narrow fibers, and other irregularly shaped structures—LCM analysis permits enrichment but not complete isolation of target proteins; samples are diluted with extraneous surrounding material. For example, when capturing even a relatively large—3 μm in diameter——inclusion body with a single 10 μm LCM spot, only ~10% of the recovered protein is from the pathology and 90% from the surrounding cellular milieu. LCM has nevertheless proven useful, however, in the proteomic analysis of systemic amyloidosis, where the amyloid deposits are very large and the feature size is typically ~50,000 μm$^2$ in area (*Sethi et al., 2012*, *2013*).

Mass spectrometry imaging and its variants can also provide spatially resolved mass spectrometry analysis by directly coupling imaging with laser ablation and achieving 1 mm–10 μm resolution (*Stoeckli et al., 2001*; *Wiseman et al., 2006*; *Wucher et al., 2007*), but it is usually used to look for specific targets rather than in discovery mode.

Recently, proximity labeling techniques such as Bio-ID and APEX have provided high spatial resolution to mass spectrometry, and these techniques have been used to elucidate the composition of several difficult to purify organelles (*Roux et al., 2012*; *Rhee et al., 2013*; *Firat-Karalar et al., 2014*; *Hung et al., 2014*). However, proximity labeling relies on genetic manipulations to express an exogenous fusion protein to label adjacent components; this protein must be specifically and accurately targeted into a particular biological structure. Imperfect localization, especially in rare structures, results in selective labeling of neighbors outside of the target region yielding false positive associations.

Many of these approaches have been brought to bear on amyloid (senile) plaques in AD brain tissue, in an attempt to identify core components, if any, in addition to Aβ. Early LCM studies utilizing 2D gel electrophoresis coupled with mass spectrometry have detected a modest number of plaque-associated proteins, including an assortment of proteins associated with cell signaling, chaperone function, membrane trafficking, and proteolysis (*Liao et al., 2004*). A more recent LCM study has suggested that amyloid plaques in AD are highly homogenous structures that are composed almost exclusively of Aβ (*Söderberg et al., 2006*). The latter study, however, employed sample washing steps with 1% SDS, which may strip away soluble plaque-associated proteins. Finally, a recent attempt to capture more plaque-associated proteins potentially lost to LCM examined the sarkosyl-insoluble fraction of brain samples from patients with AD and appropriate controls (*Gozal et al., 2009*). Altered levels of eleven specific proteins were identified by this approach but colocalization with amyloid plaques was difficult to validate in many cases—not all insoluble proteins within the brain will necessarily originate from senile plaques. A recent proteomic analysis of fractions obtained from differential centrifugation procedures that yield fractions enriched in post-synaptic proteins have demonstrated that the post-synaptic protein IRSp53 was highly down-regulated in AD (*Zhou et al., 2013*). While extant LCM and biochemical fractionation studies have identified candidate plaque-associated proteins, significant mechanistic insights have not been produced.

We have developed a semi-automated technique called spatially targeted optical microproteomics (STOMP), which combines two-photon laser scanning microscopy with photochemical affinity labeling and mass spectrometry. While two-photon excitation has previously been used to drive microscopic, 3D-resolved photochemistry (*Lee et al., 2008*), STOMP represents the first application of this technique to affinity photolabeling, offering single-micron-scale three-dimensional resolution that is an order of magnitude better than the current state of the art LCM. In addition to providing clues to the etiology of neurodegenerative disease, this new proteomic technique also has great potential to advance research in cell biology by yielding the composition of small features that are not amenable to biochemical purification.

We sought to create an unbiased discovery technique that would: (1) have high spatial resolution, (2) target any identifiable or user-defined spatial region in a biological sample, (3) forgo the need for genetic modification, (4) have sufficient sensitivity to detect minor species within the sample, and (5) have sufficient specificity such that most hits can be readily validated as being enriched in the biological structure of interest. Because STOMP does not require genetic manipulation, this technique is applicable to a large variety of biological samples that run the gamut from cell lines, primary cell cultures, ex vivo specimens, human biopsy samples to fixed post-mortem human tissue.

## Results and discussion

### The STOMP technique

In STOMP, laser light from the microscope is used not only to image the fluorescently stained specimen, but also to photochemically crosslink affinity purification ligands to protein components within pathological deposits (*Figure 1A*). The photo-affinity ligand or photo-tag used here is a peptide of sequence: 4-benzoylbenzyl-Gly-His-His-His-His-His-His-CONH$_2$ (6HisBP) (*Figure 1B*). Benzoylbenzene is a photoreactive group that forms covalent bonds with C–H and N–H groups upon excitation. The 6HisBP photo-tag molecule (molecular mass, 1105 Da) is of similar size to Congo red (molecular mass, 697 Da) and should have a similar ability to penetrate biological specimens. The specimen is fixed and permeabilized in methanol, then protein deposits in the specimen are selectively stained using antibodies or specific fluorescent dyes. The specimen is then soaked in a solution of 6HisBP to saturate the subcellular compartments with the photo-tag, and imaged using conventional confocal microscopy. The wavelength of the confocal laser is in the visible range and does not excite the UV-absorbing photo-tag. Optical sectioning by confocal microscopy is used to collect a 3D image of the protein deposits in the specimen. This image in turn is used to generate a list of spatial coordinates—a 'mask' file—identifying every point in the tissue containing pathological protein deposits (*Figure 1C*).

The photo-tagging procedure uses two-photon microscopy (*Skoch et al., 2006*) rather than conventional confocal microscopy. Two-photon microscopy can excite a small volume (<1 μm$^3$, *Figures 1A* and *Figure 2*) anywhere in the specimen, while conventional confocal microscopy excites a continuous cone that spans the entire thickness of the specimen, which would result in the covalent attachment of the photo-tag to off-target regions. A custom software macro was written that reads the mask file and selectively delivers two-photon excitation light to each protein deposit present, while leaving the remainder of the specimen untouched by excitation light (the custom macro is available as *Source code 1*). The STOMP macro provides a semi-automated tool for photo-tagging proteins in pathological deposits. Importantly, unlike LCM, no manual tracing of features is required with STOMP.

We rely on reversible binding of photo-tagged proteins using nickel affinity chromatography. Histidine-rich proteins ubiquitous in all mammalian tissues represent potential contaminants that could bind the nickel affinity beads and confound the STOMP analysis. To circumvent this problem, the first step of the procedure involves limited treatment of the tissue sections with dilute diethyl pyrocarbonate (DEPC). DEPC covalently modifies histidyl residues that abolishes their ability to bind nickel (*Wallis and Holbrook, 1973*).

After completion of the photo-tagging step, the specimen is solubilized in buffer containing 2% SDS, 8 M urea, and β-mercaptoethanol (β-ME). The photo-tagged proteins are purified using nickel affinity beads that bind the hexahistidyl moiety of 6HisBP. The purified proteins are then analyzed by gel electrophoresis (*Figure 1D*) and identified by database searching spectra from peptides resulting from digestion and liquid-chromatography and tandem mass spectrometry (LC-MS/MS).

### The resolution of STOMP

To establish the spatial resolution limits of our photo-tagging procedure, we tagged single-voxel spots within sections of methanol-fixed murine brain tissue. By subsequently staining the tissue with an anti-hexahistidine antibody, we were able to use confocal immunofluorescence imaging to directly visualize and measure the extent of the photo-tagged volume.

We measured the diameter (full width at half-maximum, FWHM) of the phototagged volume at 0.67 μm along the *x* and *y* axes, and 1.48 μm along the *z* axis (*Figure 2*). Taking the excited region to be an ellipsoid, the total volume of a single spot is 0.38 μm$^3$.

### STOMP analysis of amyloid plaques in a transgenic mouse model of AD

We used TgCRND8 mice, a well-characterized transgenic mouse model of AD (*Chishti et al., 2001*), as a model system for the development of the STOMP technique. These mice express a human form of the amyloid precursor protein carrying two mutations associated with familial AD, and they produce amyloid plaques and exhibit spatial learning impairments by 3 months of age. This study used frozen

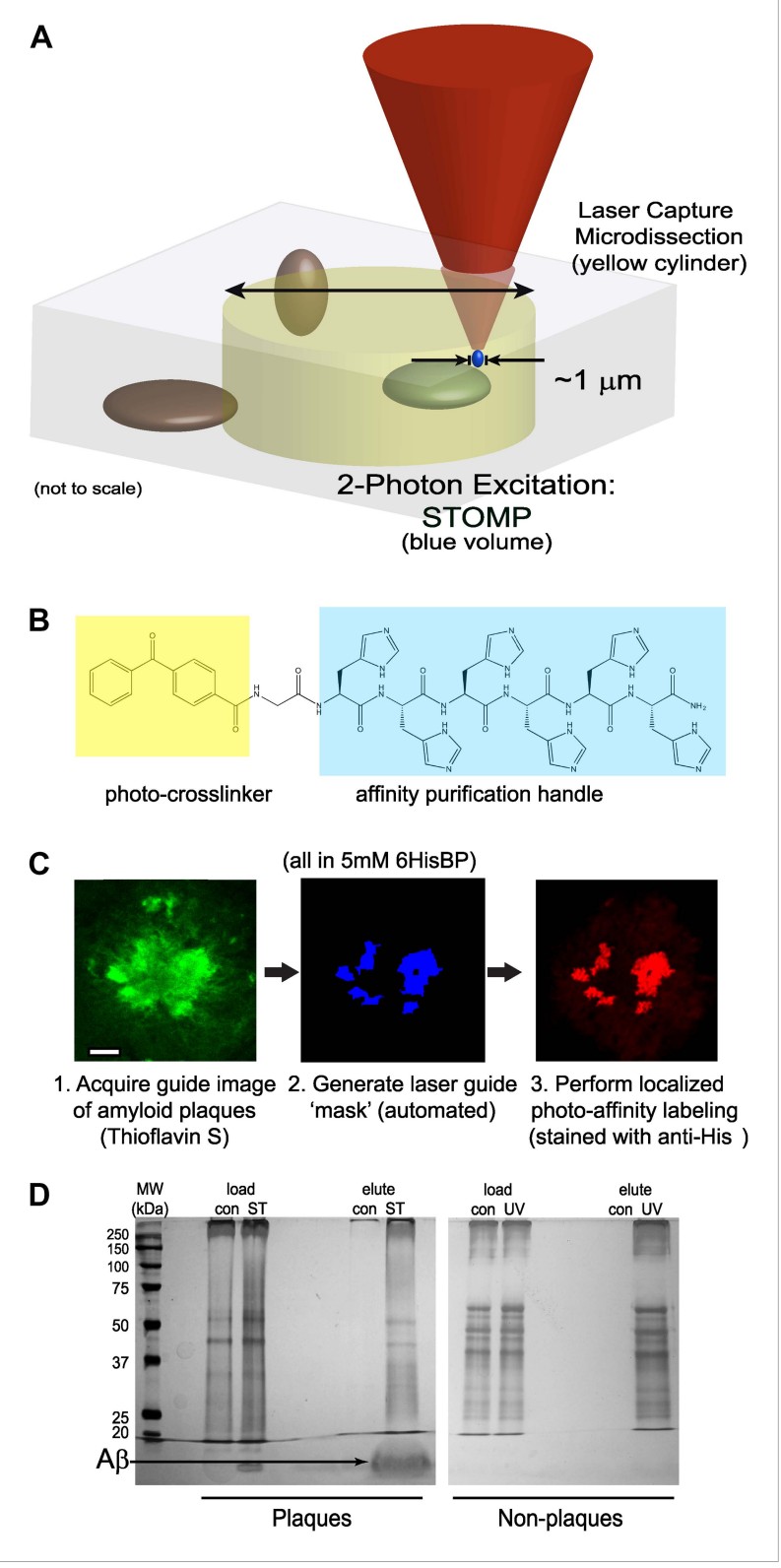

**Figure 1**. Overview of STOMP technology. (**A**) Schematic diagram comparing volumes selected for excision by laser capture microdissection (LCM) (yellow cylinder) and spatially targeted optical microproteomics (STOMP) (blue volume). The STOMP excitation volume is limited approximately to the point spread function of a high numerical aperture lens (<1 µm in the xy plane, ~1.0 µm axially) compared to ~10 µm for LCM. (**B**) Structure of the bifunctional

*Figure 1. continued on next page*

*Figure 1. Continued*

photocrosslinker used in this study; the affinity purification hexahistidine peptide is highlighted in blue, the photocrosslinker benzophenone group is highlighted in yellow. (C) Overview of the STOMP protocol. Tissue sections or cells are first fixed in cold methanol and stained using specific dyes or antibodies. (1) A guide image is acquired at a wavelength that does not activate the photocrosslinker. (2) The guide image is converted to a digital mask that directs the two-photon laser. (3) Two-photon laser selectively illuminates region of interest. The final image shown is anti-His tag immunofluorescence corresponding to areas of photo-tag crosslinking. (D) Silver stain of SDS-PAGE of material retrieved from selective STOMP of CRND8 mouse plaques (left panel) or non-plaque regions of an adjacent brain section (right panel). In each case the control ('con') is non-specific binding to $Ni^{2+}$-NTA agarose beads. Load is 1% of total protein after solubilization of tissue. Note that the predominant protein photo-tagged in the STOMP sample is Aβ (arrow), which is not visible in the non-plaque experiment because of its low overall abundance.

sections (post-fixed in methanol) of the brains known to contain plaques, from TgCRND8 mice of 8 months of age.

Sets of serial sections on separate slides were treated with DEPC, stained with ThS, and soaked in a solution of 6HisBP. Slides were imaged by confocal microscopy to identify ThS-positive amyloid

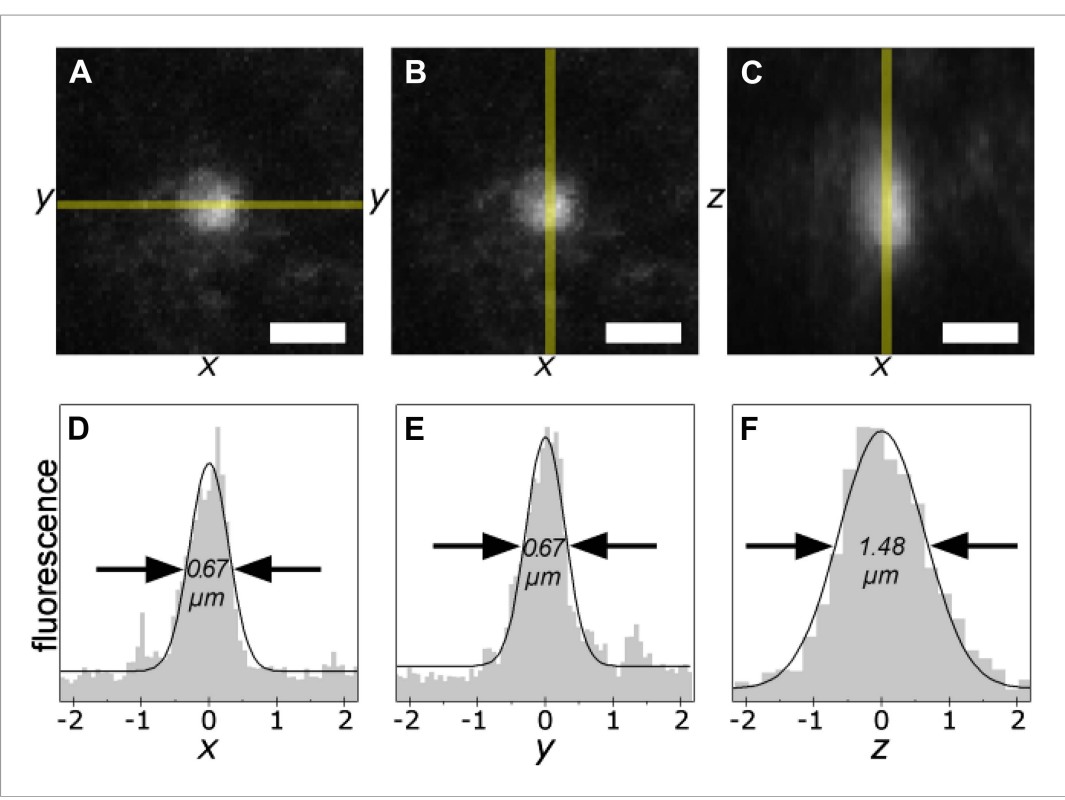

**Figure 2**. The smallest photo-tagging volume is less than 0.5 µm³. A single voxel in a TgCRND8 tissue section was photo-tagged. The volume of photo-excitation was measured by confocal fluorescence imaging of the resulting photo-tagged spot. Maximum-intensity projections of the xy- (**A**, **B**) and xz- (**C**) planes are shown. Scale bar 1 µm. Fluorescence profiles of the indicated regions (shaded yellow) are shown (**D–F**). The width of the peaks is 0.67 µm in x and y, and 1.48 µm along the z axis, corresponding to an ellipsoidal excited volume of 0.38 µm³.

The following figure supplements are available for figure 2:

**Figure supplement 1**. Cross correlation of technical replicates.

**Figure supplement 2**. Correlation of biological replicates.

deposits. Confocal images of ThS-positive amyloid deposits (*Figure 1C1*) were used to construct individual masks (*Figure 1C2*). Because our technique hinges on selective photolabeling and purification, we needed to assess the extent of non-specific labeling of 6HisBP in ambient light and under immunofluorescence excitation, as well as non-specific binding to affinity purification beads. Adjacent sections were put aside as 'dark' controls used to assess the extent of non-specific labeling of 6HisBP to proteins caused by confocal laser light (488 nm) exposure or other handling, and to assess nonspecific binding of proteins to nickel affinity beads.

The STOMP macro was used to deliver two-photon excitation light to regions of the specimen corresponding to each pixel in the mask image. This excitation light has two effects. First, and most importantly, it photo-activates 6HisBP molecules that are in the amyloid deposits causing photo-tagging of constituent proteins. Second, it serendipitously photo-bleaches the ThS fluorophores present in regions of the specimen targeted by the mask. Immunofluorescence staining of the tissue section with anti-His$_6$ antibody after photo-activation superimposes on the digital mask, thus highlighting the very high accuracy of targeting of the two-photon laser (*Figure 1C3*). STOMP combines microscopy with selective photo-labeling to accurately resolve, capture, and affinity-label highly irregular shaped micron-scale structures, via a semi-automated procedure.

After solubilization of the specimen, the photo-tagged proteins were bound to nickel affinity beads. Each sample was divided into two portions: one used for mass spectrometry (*Table 1*) and one for gel electrophoresis and silver staining (*Figure 1D*). The dark control sample, which—aside from two-photon excitation—was treated identically to the STOMP sample, was run alongside the STOMP sample. It shows very few bands in the silver-stain gel from material bound to the nickel-nitrilotriacetic acid (Ni-NTA) beads compared to the STOMP sample, confirming that nonspecific photo-tagging and nonspecific binding of proteins to the nickel affinity beads is minimal. In addition to a number of proteins ranging in molecular weight from 20 kDa to >250 kDa, the STOMP sample contains large amounts of a low molecular weight protein that was subsequently identified as Aβ (4.5 kDa) (*Figure 1D*).

As an additional control, an entire brain section fixed in methanol and soaked with photo-tag was photo-activated by exposure to 365 nm ultraviolet light. Section-wide photo-activation of this specimen caused indiscriminate photo-tagging of proteins in the specimen. Gel electrophoresis of the indiscriminately photo-tagged proteins reveals a very different pattern of protein bands compared to the specimen in which amyloid plaques were specifically targeted for STOMP analysis (*Figure 1D*).

## Identification of photo-tagged amyloid plaque proteins by mass spectrometry

The total volume of tissue that needs to be photo-tagged to obtain sufficient material for mass spectrometry analysis is a subjective matter and also depends on instrument sensitivity, fixation method used, sample complexity and perhaps other factors. Greater amounts of tissue will increase sensitivity and enable identification of lower abundance proteins, particularly in very complex samples containing a large number of unique proteins. In this study, we were able to identify proteins present in relative abundance in the pathological feature being examined using a total photo-tagged volume of approximately $2 \times 10^6$ μm$^3$. (This corresponds to a photo-tagged region of 500 μm × 500 μm in an 8-μm thick tissue section or a cube of photo-tagged tissue 130 μm on a side.) Lowering the stringency will increase the number of hits but may introduce more false positives. Two frozen sections of TgCRND8 brain tissue were sufficient to produce the necessary volume of photo-tagged tissue required for one STOMP experiment, and the process required a total of 12–16 hr of microscope time per sample (6–8 hr per section).

Based on our determination of the resolution of STOMP, regions of interest (ROIs) as small as 0.38 μm can be analyzed (*Figure 2*). To analyze such small features, however, sufficient numbers of ROIs must be captured, and this may necessitate pooling photo-tagged material from several tissue sections.

Protein identification via mass spectrometry involved on-bead digestion of the photo-tagged proteins with endoproteinase Lys-C followed by trypsin and LC-MS/MS analysis. We performed technical and biological replicates of the proteomic analyses of our STOMP experiment. The technical replicates (*Figure 2—figure supplement 1*) demonstrate a very high degree of correlation ($r^2 = 0.98$), and the biological replicates demonstrate significant overlap in proteins identified (*Figure 2—figure supplement 2*). The replicate studies illustrate the high reproducibility of the technique. The list of all

**Table 1**. Proteins statistically significantly enriched in the amyloid plaques of TgCRND8 mouse brain identified and retrieved by STOMP

| Protein | Uniprot ID | STOMP counts | SAINT score | Previous reports of enrichment in plaques (*Söderberg et al., 2006*) |
|---|---|---|---|---|
| Amyloid beta A4 protein | P12023 | 378 | 0.83 | Detected, enriched |
| Synaptosomal-associated protein 25 | P60879 | 88 | 1.00 | Detected, not enriched |
| Cytochrome c1 | Q9D0M3 | 70 | 0.98 | Detected, not enriched |
| Excitatory amino acid transporter 2 | P43006 | 64 | 1.00 | Detected, not enriched |
| V-type proton ATPase subunit B | P62814 | 56 | 0.80 | Detected, enriched |
| Vesicle-associated membrane protein 2 | P63044 | 52 | 0.87 | Detected, not enriched |
| Pyruvate kinase isozymes M1/M2 | P52480 | 49 | 0.98 | Detected, not enriched |
| Fructose-bisphosphate aldolase A | P05064 | 47 | 1.00 | Previously unreported |
| Cytochrome b-c1 complex subunit 1 | Q9CZ13 | 47 | 0.92 | Previously unreported |
| Guanine nucleotide-binding protein G(o) subunit alpha | P18872 | 41 | 0.98 | Previously unreported |
| Cytochrome c oxidase subunit 2 | P00405 | 40 | 0.97 | Detected, not enriched |
| Tubulin alpha-1B chain | P05213 | 39 | 0.98 | Detected, not enriched |
| 4-aminobutyrate aminotransferase | P61922 | 39 | 0.81 | Previously unreported |
| Tubulin beta-3 chain | Q9ERD7 | 38 | 1.00 | Detected, not enriched |
| V-type proton ATPase catalytic subunit A | P50516 | 37 | 0.99 | Detected, enriched |
| Aralar 1 | Q8BH59 | 37 | 1.00 | Previously unreported |
| Citrate synthase, mitochondrial | Q9CZU6 | 37 | 1.00 | Detected, not enriched |
| Clathrin heavy chain 1 | Q68FD5 | 35 | 1.00 | Detected, enriched |
| Alpha-internexin | P46660 | 30 | 0.98 | Previously unreported |
| NADH dehydrogenase 1 alpha subcomplex subunit 9, mitochondrial | Q9DC69 | 30 | 0.98 | Detected, not enriched |
| Fructose-bisphosphate aldolase C | P05063 | 28 | 0.91 | Detected, not enriched |
| Synapsin-2 | Q64332 | 27 | 0.89 | Detected, not enriched |
| Dynamin-1 | P39053 | 26 | 0.99 | Detected, not enriched |
| NADH dehydrogenase 1 alpha subcomplex subunit 10, mitochondrial | Q99LC3 | 24 | 0.89 | Detected, not enriched |
| Spectrin alpha chain, brain | P16546 | 22 | 1.00 | Detected, not enriched |
| Succinate dehydrogenase flavoprotein subunit, mitochondrial | Q8K2B3 | 21 | 0.99 | Detected, not enriched |
| Synapsin-1 | O88935 | 20 | 0.97 | Detected, not enriched |
| Rab GDP dissociation inhibitor alpha | P50396 | 20 | 0.94 | Detected, not enriched |
| V-type proton ATPase 116 kDa subunit a | Q9Z1G4 | 19 | 0.98 | Detected, enriched |
| Hexokinase-1 | P17710 | 18 | 1.00 | Detected, not enriched |
| Heat shock protein HSP 90-alpha | P07901 | 18 | 0.93 | Detected, enriched |
| Ubiquitin thioesterase OTUB1 | Q7TQI3 | 18 | 0.85 | Previously unreported |
| Vesicle-fusing ATPase | P46460 | 17 | 0.88 | Previously unreported |
| Spectrin beta chain, brain 1 | Q62261 | 17 | 1.00 | Detected, not enriched |
| NADH-ubiquinone oxidoreductase 75 kDa subunit, mitochondrial | Q91VD9 | 16 | 0.95 | Detected, not enriched |
| Neurofilament light polypeptide | P08551 | 15 | 0.85 | Detected, enriched |
| Neurochondrin | Q9Z0E0 | 15 | 0.97 | Detected, not enriched |
| Heat shock protein HSP 90-beta | P11499 | 14 | 0.98 | Detected, enriched |
| Na/K-transporting ATPase subunit alpha-2 | Q6PIE5 | 14 | 0.95 | Detected, not enriched |

*Table 1. Continued on next page*

*Table 1. Continued*

| Protein | Uniprot ID | STOMP counts | SAINT score | Previous reports of enrichment in plaques (*Söderberg et al., 2006*) |
|---|---|---|---|---|
| Excitatory amino acid transporter 1 | P56564 | 14 | 0.93 | Detected, not enriched |
| Microtubule-associated protein 6 | Q7TSJ2 | 13 | 0.87 | Detected, not enriched |
| Serine/threonine-protein phosphatase 2A 65 kDa regulatory subunit A alpha isoform | Q76MZ3 | 11 | 0.80 | Detected, not enriched |
| Catenin beta-1 | Q02248 | 11 | 0.83 | Detected, not enriched |
| Tenascin-R | Q8BYI9 | 10 | 0.85 | Detected, not enriched |
| Alpha-actinin-1 | Q7TPR4 | 10 | 0.98 | Detected, not enriched |
| Ubiquitin-like modifier-activating enzyme 1 | Q02053 | 9 | 0.93 | Detected, not enriched |
| Pyruvate carboxylase, mitochondrial | Q05920 | 9 | 0.92 | Previously unreported |
| Ankyrin-2 | Q8C8R3 | 2 | 0.85 | Detected, not enriched |

Table is sorted in descending order of abundance, by normalized spectral counts.

identified proteins in amyloid plaques of TgCRND8 mice based on our high stringency criteria is summarized in *Table 1*.

The aim of STOMP is to provide a compendium of annotated proteins present within the ROI under study—whether or not they are enriched or depleted relative to the surrounding tissue. As regards the quantitative nature of the data, *Table 1* collects and summarizes our high-confidence mass spectrometry protein identifications, sorted by molecular-weight-normalized spectral counts: a semi-quantitative proxy for protein abundance. As expected, the most abundant protein found in STOMP from TgCRND8 mouse plaques was Aβ (*Figure 1D*, *Table 1*). In addition to Aβ, we identified 454 proteins in at least one of two biological replicates (*Supplementary file 1*). Combining our biological and technical replicates with our negative controls ('dark' samples and replicates), we used the Significance Analysis of INTeractome (SAINT) algorithm (*Choi et al., 2011*), which uses Bayesian inference to assign a probability of a true interaction, using Aβ as the bait protein. We found 62 proteins with probability >0.8 of being found associated with the plaques, and 146 with probability >0.5 (*Supplementary file 1*). Among these, synaptosomal proteins were highly enriched, including SNAP25, vesicle-associated membrane protein 2 (VAMP2), vesicular proton pump ATPase (v-ATPase), and synapsins 1 and 2. Apolipoprotein E, which is sometimes associated with senile plaques (*Thal et al., 1997*), was also detected, albeit at a SAINT score of 0.70 (*Supplementary file 1*). Certain common synaptic markers—including chromogranin A/C, synaptophysin, synaptogyrin I, synapto-porin, α-synuclein, and VGLUT1 and 2—were not detected in the plaques. Interestingly, the plaques were also devoid of uniquely post-synaptic proteins. We also identified many mitochondrial proteins; although the reason for this is unclear, it may be that the intrinsic hydrophobicity of mitochondrial membrane proteins may predispose them to associate with amyloid plaques. Comparison with proteins found from STOMP analysis of non-plaque areas revealed only limited overlap (*Supplementary file 2*); VAMP2, synapsin 1, and dynamin 1 are present in both plaque and non-plaque STOMP analysis but are significantly greater enriched in the plaque STOMP samples.

## Validation of the STOMP results in TgCRND8 mice with immunofluorescence and immunohistochemistry

Using confocal microscopy, we examined colocalization of ThS-positive plaques with immunfluor-escently labeled Aβ, SNAP25, VAMP2, v-ATPase, synapsin 1, and ApoE. All were found to be present and are apparently genuine components of the amyloid plaque (*Figure 3*). Optical sectioning demonstrates the presence of each of these proteins in the core of the amyloid plaques suggesting they were incorporated at the earliest stages of plaque formation. Staining of SNAP25, VAMP2, and v-ATPase reveal an immunopositive halo surrounding each of the plaques indicating an accumulation of these proteins not only in the amyloid core but also in the vicinity of the plaques; interestingly, synapsin 1 immunostaining did not have this halo.

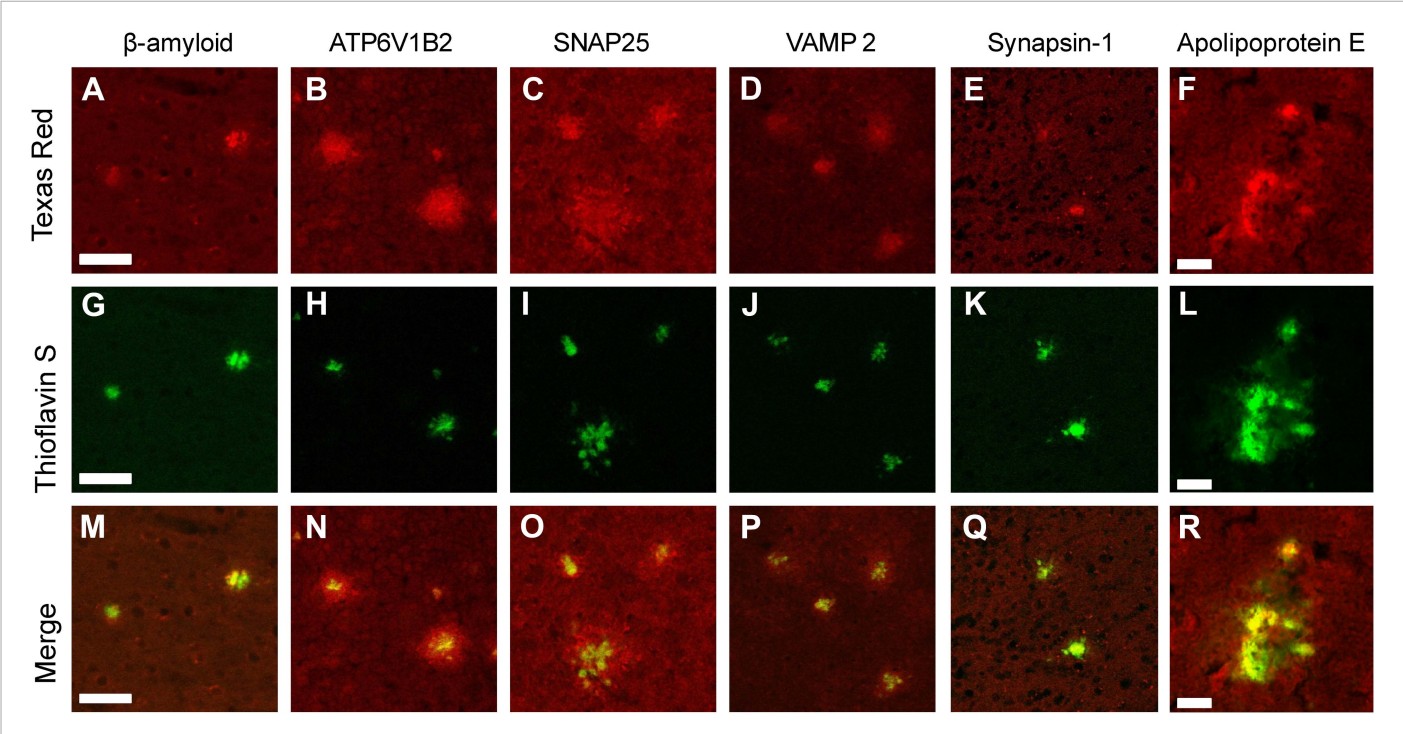

**Figure 3**. Immunofluorescent confirmation of synaptic proteins in amyloid plaques of TgCRND8 mice. Colocalization of ThS-stained plaques with beta-amyloid positive control (**A**, **G**, **M**) and the synaptic proteins ATP6V1B2 (**B**, **H**, **N**), SNAP25 (**C**, **I**, **O**), VAMP2 (**D**, **J**, **P**), synapsin 1 (**E**, **K**, **Q**), and ApoE (**F**, **L**, **R**). ATP6V1B2, SNAP25, and VAMP2 (**N**–**P**) all show a halo of elevated protein concentration in the region surrounding the dense core of the plaque. Scale bar 50 μm, 20 μm in the ApoE panels.

As mentioned above, certain common synaptic markers were not detected by the STOMP analysis. Immunostaining of some of these markers, syntaxin 1, synaptophysin, and α-synuclein, confirmed that these proteins are not concentrated in the amyloid plaques (*Figure 4*). α-Synuclein has been reported to be associated with senile plaques in human patients with AD; however, TgCRND8 mice are known to not accumulate α-synuclein in their plaques (*Xu et al., 2002*).

It is not clear why the amyloid deposits contain some pre-synaptic proteins, but not others, and are devoid of post-synaptic proteins. One possibility is that damaged dystrophic neurites that are known to surround the plaques (*Chishti et al., 2001*) are spilling their contents and the physical properties of certain pre-synaptic proteins cause them to bind with especially high affinity to amyloid plaques. In this regard, it is noteworthy that many of these proteins are membrane proteins with coiled coil domains (*Takamori et al., 2006*). Another possibility, which is not mutually exclusive and provides an explanation for the presence of pre-synaptic proteins in plaques, is that Aβ fibrils and/or oligomers bind directly to synaptic vesicles and interfere with synaptic function. Evidence for a damaging effect of Aβ on synapses and dendrites has been available for some time (*Mattson et al., 1998*). Diffusible Aβ oligomers added to cultured primary neurons have been shown to associate with synaptosomes and a concomitant synaptic deterioration is observed (*Lacor et al., 2007*). In post-mortem AD tissue and in transgenic AD mice, an inverse correlation between levels of Aβ oligomers and levels of synaptic proteins was noted (*Pham et al., 2010*). The reduction in the levels of detectable synaptic protein in AD tissue may be the result of the aggregation and entrapment of the synaptic proteins in the amyloid plaques. In sum, the STOMP technique has provided clear evidence of interaction of amyloid plaques with pre-synaptic proteins.

## STOMP analysis of senile plaques from post-mortem AD brain

After developing the STOMP method using brain sections of TgCRND8 mice as the test subject, we applied the technique to senile plaques from a case of AD. Human tissue is inherently more difficult to

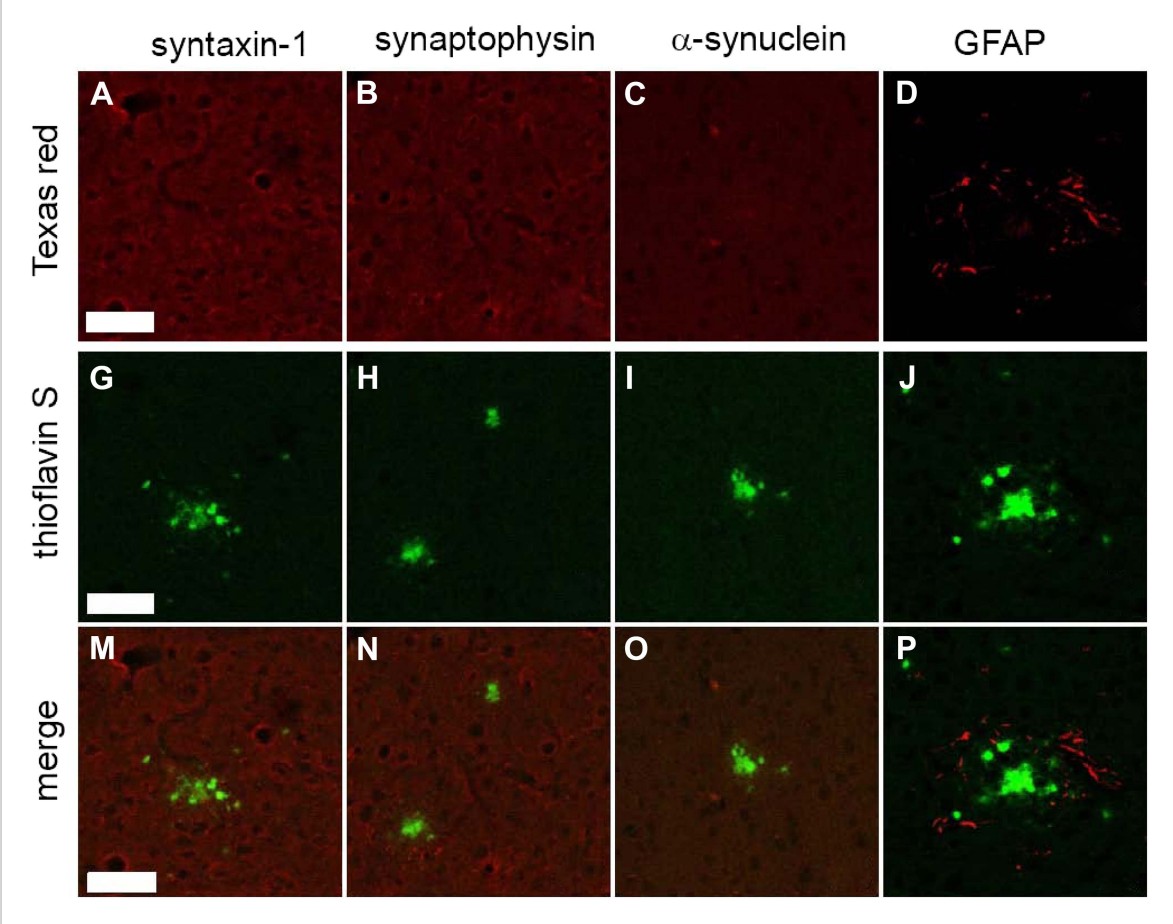

**Figure 4**. Common synaptic or disease-associated proteins in plaques of TgCRND8 mice not detected by STOMP are also absent by immunofluorescence. The SNARE protein syntaxin-1 (**A**), synaptic marker synaptophysin (**B**), the neuronal protein alpha-synuclein (**C**) are absent from ThS-positive plaques (**G**, **H**, **I**, **M**, **N**, **O**). Staining of the glial protein GFAP (**D**) surrounds the ThS-positive plaques but does not infiltrate the plaque core (**J**, **P**). Scale bar 50 μm.

work with than animal tissue because of biochemical changes associated with post-mortem degradation. Furthermore, formalin, the fixative of choice for human tissue, causes covalent modification of proteins and confounds mass spectrometry analysis. These limitations notwithstanding, we find the STOMP technique is suitable for the analysis of formalin-fixed post-mortem human tissue. Senile plaques from brain sections of a severe case of AD were visualized with ThS and photo-tagged using a procedure identical to that used for the TgCRND8 tissue. Mass spectrometry analysis of the protein composition of these senile plaques identified 60 proteins that were statistically significantly enriched, with Aβ being the most abundant protein component (*Table 2*). Synaptic proteins and proteins involved in synaptic vesicle transport comprised 15% of the abundant protein component of the plaques. For validation of the STOMP analysis of senile plaques, immunofluorescence staining was employed, revealing that GFAP and SNAP25 were detected by both immunofluorescence, while tau and α-synuclein were not detected by either technique (*Figure 5*).

Gliosis is a well-known feature of AD, and reactive glial cells are often found surrounding amyloid plaques (*Mandybur and Chuirazzi, 1990*; *Sofroniew and Vinters, 2010*). Glial proteins, GFAP and excitatory amino acid transporter (EAAT) (*Anderson and Swanson, 2000*), were both detected in (human) senile plaques (*Table 2*); however, only EAAT was detected in TgCRND8 amyloid plaques (*Table 1*). This apparent discrepancy is likely due to astrocytes being more intimately associated with amyloid plaques in human tissue (*Figure 5A,E,I*) compared to plaques in the TgCRND mouse (*Figure 4D,J,P*). Tau protein was also not among the abundant plaque proteins.

**Table 2.** Proteins statistically significantly enriched in senile plaques from a patient with AD identified and retrieved by STOMP

| Protein | Uniprot ID | STOMP counts | Dark counts | Previous reports of enrichment in plaques (*Söderberg et al., 2006*) |
|---|---|---|---|---|
| Amyloid beta A4 protein | P05067 | 898 | 0.00 | Detected, enriched |
| Tubulin alpha-1A chain | Q71U36 | 39.4 | 1.99 | Detected, not enriched |
| Tubulin beta-2A chain | Q13885 | 23.0 | 3.51 | Detected, not enriched |
| Actin, cytoplasmic 1 | P60709 | 13.8 | 0.00 | Detected, not enriched |
| Glial fibrillary acidic protein | P14136 | 11.0 | 0.00 | Detected, enriched |
| Alpha-internexin | Q16352 | 10.8 | 0.90 | Previously unreported |
| Synaptosomal-associated protein 25 | P60880 | 9.65 | 0.00 | Detected, not enriched |
| Carbonyl reductase [NADPH] 1 | P16152 | 9.05 | 0.00 | Detected, not enriched |
| ATP synthase subunit beta, mitochondrial | P06576 | 9.28 | 0.00 | Detected, enriched |
| Tubulin polymerization-promoting protein | O94811 | 6.33 | 0.00 | Previously unreported |
| Tubulin beta-3 chain | Q13509 | 5.95 | 0.00 | Previously unreported |
| Cytochrome b-c1 complex subunit 8 | O14949 | 5.05 | 0.00 | Previously unreported |
| Calcium/calmodulin-dependent protein kinase type I | Q9UQM7 | 5.08 | 0.00 | Previously unreported |
| Immunoglobulin superfamily member | Q969P0 | 4.23 | 0.00 | Previously unreported |
| V-type proton ATPase subunit B, kidney isoform | P15313 | 3.96 | 0.00 | Detected, enriched |
| Vesicle-associated membrane protein 2 | P63027 | 3.95 | 0.00 | Detected, not enriched |
| Pyruvate dehydrogenase E1 component subunit beta | P11177 | 3.82 | 0.00 | Detected, not enriched |
| Fructose-bisphosphate aldolase C | P09972 | 3.80 | 0.00 | Detected, not enriched |
| Ferritin light chain | P02792 | 3.75 | 0.00 | Detected, not enriched |
| Neurofilament heavy polypeptide | P12036 | 3.33 | 0.89 | Detected, not enriched |
| Cytochrome c1, heme protein, mitochondrial | P08574 | 2.82 | 0.00 | Detected, not enriched |
| Peptidyl-prolyl cis-trans isomerase | Q13526 | 2.74 | 0.00 | Detected, not enriched |
| Phosphoglycerate mutase 1 | P18669 | 2.60 | 0.00 | Detected, not enriched |
| Beta-actin-like protein 2 | Q562R1 | 2.38 | 0.00 | Detected, not enriched |
| Creatine kinase B-type | P12277 | 2.34 | 0.00 | Detected, not enriched |
| Transketolase | P29401 | 2.21 | 0.00 | Detected, not enriched |
| Alpha-enolase | P06733 | 2.12 | 0.00 | Detected, not enriched |
| Excitatory amino acid transporter 1 | P43003 | 2.10 | 0.00 | Previously unreported |
| 40S ribosomal protein S8 | P62241 | 2.07 | 0.00 | Previously unreported |
| 60 kDa heat shock protein, mitochondrial | P10809 | 2.05 | 0.00 | Detected, not enriched |
| Stress-70 protein, mitochondrial | P38646 | 2.04 | 0.00 | Detected, not enriched |
| Protein SERCA1 | Q96JX3 | 2.02 | 0.00 | Previously unreported |
| Spectrin beta chain, brain 1 | Q01082 | 1.82 | 0.00 | Detected, enriched |
| Fascin | Q16658 | 1.83 | 0.00 | Detected, not enriched |
| 6-phosphogluconolactonase | O95336 | 1.82 | 0.00 | Previously unreported |
| Thioredoxin-dependent peroxide reductase | P30048 | 1.81 | 0.00 | Detected, not enriched |
| Glutamine synthetase | P15104 | 1.78 | 0.00 | Previously unreported |
| Clathrin heavy chain 1 | Q00610 | 1.70 | 0.00 | Detected, not enriched |

*Table 2. Continued on next page*

*Table 2. Continued*

| Protein | Uniprot ID | STOMP counts | Dark counts | Previous reports of enrichment in plaques (*Söderberg et al., 2006*) |
|---|---|---|---|---|
| Elongation factor Tu, mitochondrial | P49411 | 1.51 | 0.00 | Detected, not enriched |
| Tubulin alpha-4A chain | P68366 | 1.50 | 0.00 | Detected, not enriched |
| Methionine adenosyltransferase 2 subunit beta | Q9NZL9 | 1.33 | 0.00 | Previously unreported |
| Dihydropyrimidinase-related protein 4 i | O14531 | 1.21 | 0.00 | Previously unreported |
| Tenascin-R | Q92752 | 1.17 | 0.00 | Detected, not enriched |
| Microtubule-associated protein 6 | Q96JE9 | 1.16 | 0.29 | Detected, not enriched |
| Prelamin-A/C | P02545 | 1.01 | 0.00 | Previously unreported |
| Neurofascin | O94856 | 1.00 | 0.00 | Detected, not enriched |
| Protein kinase C and casein kinase substrate | Q9BY111 | 0.98 | 0.00 | Detected, not enriched |
| Hexokinase-1 | P19367 | 0.98 | 0.00 | Detected, not enriched |
| NADH-ubiquinone oxidoreductase 75 kDa subunit | P283311 | 0.94 | 0.00 | Detected, not enriched |
| Coronin-1C | Q9ULV4 | 0.94 | 0.00 | Detected, enriched |
| Fumarate hydratase, mitochondrial | P07954 | 0.91 | 0.00 | Previously unreported |
| Serine/threonine-protein kinase PAK 1 | Q13153 | 0.82 | 0.00 | Previously unreported |
| V-type proton ATPase 116 kDa subunit a isoform 1 | Q93050 | 0.78 | 0.00 | Detected, enriched |
| Contactin-associated protein 1 | P78357 | 0.64 | 0.00 | Detected, not enriched |
| Ubiquitin-like modifier-activating enzyme 1 | P22314 | 0.64 | 0.00 | Previously unreported |
| ATP-citrate synthase | P53396 | 0.62 | 0.00 | Detected, not enriched |
| Heat shock protein HSP 90-alpha | P07900 | 0.59 | 0.00 | Detected, enriched |
| Aconitate hydratase, mitochondrial | Q99798 | 0.58 | 0.00 | Detected, not enriched |
| Spectrin alpha chain, brain | Q13813 | 0.44 | 0.00 | Detected, not enriched |
| Microtubule-associated protein 1B | P46821 | 0.37 | 0.00 | Detected, not enriched |

Table is sorted in descending order of abundance, by normalized spectral counts.

Immunofluorescence staining shows that tau, which is contained in dystrophic neurites that surround the plaques, do not occupy the same physical space as the plaques and were hence not detected in the STOMP analysis (*Figure 3B,H,N*). SNAP25, on the other hand, was detected in the STOMP analysis and was seen to colocalize with senile plaques (*Figure 5D,H,L*) and amyloid plaques from TgCRND8 mice (*Figure 3C,I,O*). α-synuclein is part of the NAC (non-amyloid components) of plaques—with a possible role in fibril formation but not part of the plaque core (*Culvenor et al., 1999*; *Wirths and Bayer, 2003*). These proteins were found not to be enriched in the plaques of this case of AD as detected by both STOMP analysis (*Table 2*) or by immunofluorescent staining (*Figure 5C,G,K*).

Immunohistochemical staining of pre-synaptic proteins, SNAP25, VAMP2, and synaptophysin in senile plaques from human patients with AD was similar to the immunofluoresence staining in the AD mice, with some notable differences (*Figure 6*). Senile plaques in the human AD tissue were positive for both SNAP25 and VAMP2. However, synaptophysin staining, which was absent in the amyloid deposits of AD mice, displayed a unique punctate lobular profile that decorated the periphery of the plaque in the human tissue (*Figure 6*). These STOMP analyses demonstrate the molecular differences of amyloid deposits in AD mice compared to senile plaques in AD patients, but confirm the segregation of these pre-synaptic proteins in and around amyloid plaques and point to their potential importance in AD pathophysiology.

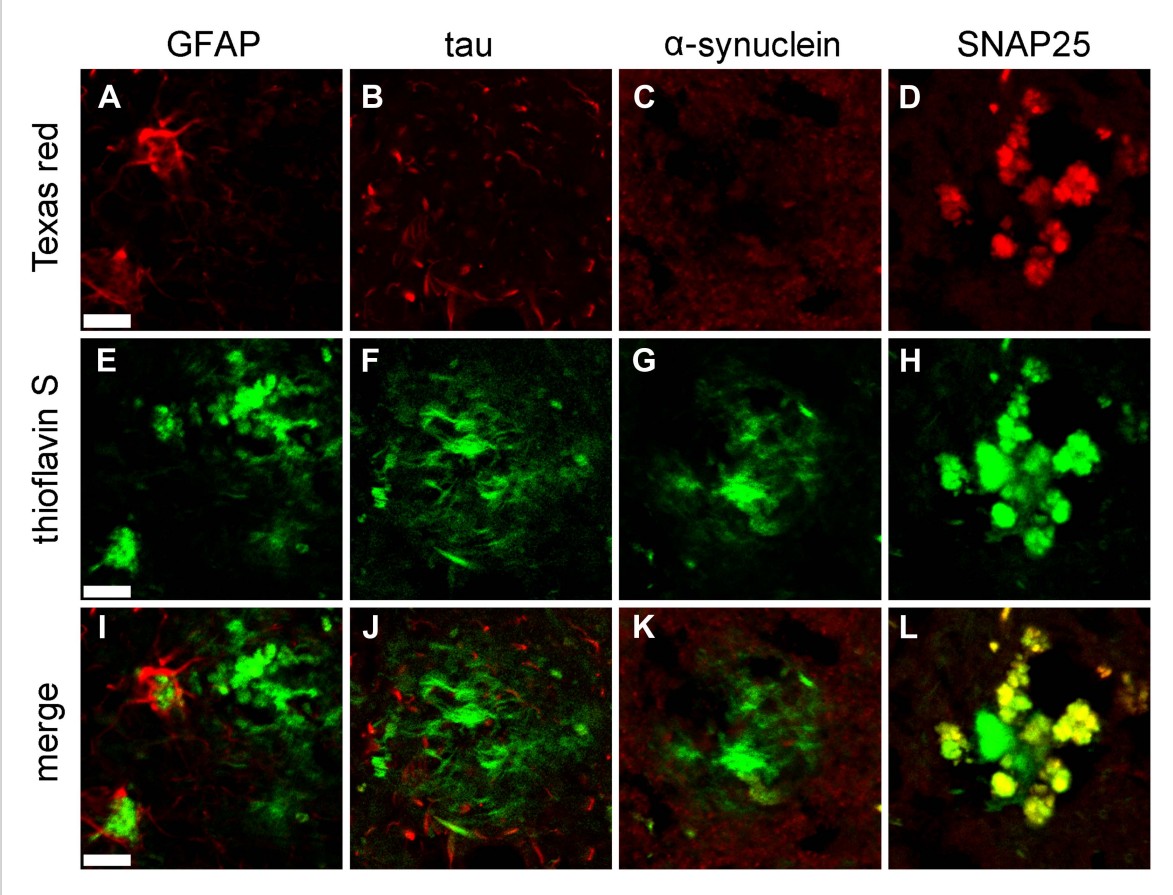

**Figure 5**. Immunofluorescent confirmation of results of STOMP analysis of senile plaques in a case of AD. Immunofluorescent confirmation of results of STOMP analysis of senile plaques in a case of AD. GFAP (**A**), a marker of glial cells, surrounds the ThS-positive plaques and partially infiltrates the plaque core (**E**, **I**). Tau in dystrophic neurites (**B**) surrounds the ThS-positive plaques but does not infiltrate the plaque core (**F**, **J**). α-synuclein (**C**) is absent from ThS-positive plaques (**G**, **K**). SNAP25 colocalizes with ThS-positive plaques in thiscase of AD (**D**, **H**, **L**). Scale bar is 20 µm.

## Comparison with previously published data

Some proteins identified by STOMP (including v-ATPase, dynamin, SNAP25, VAMP2, synapsin, clathrin, ankyrin, and so on; see *Tables 1, 2* and *Supplementary file 3*) were detected in previous LCM-based studies; however, with the exception of v-ATPase and dynamin, these proteins were not previously shown to be significantly or sufficiently enriched compared to the control samples (*Liao et al., 2004*). The highly specific nature of STOMP protein labeling and recovery permits very sensitive detection of proteins present in plaques, even if those proteins are also abundant in the remainder of the brain.

## Conclusions

We have shown that the STOMP technique can be used to determine the proteomic composition of micron-scale microscopic features. The procedure does not require large quantities of tissue and can be completed in a reasonable amount of time. Using amyloid plaques in TgCRND8 mice as a test subject, we demonstrated that STOMP is capable of determining the protein composition of pathological deposits in neurodegenerative disease. Our discovery that there is an accumulation of certain pre-synaptic proteins in AD plaques and in TgCRND8 mice provides direct evidence for an association between amyloid plaque formation and synaptic function. While our initial work in characterizing amyloid deposits in mouse brain used methanol fixed tissues to minimize problems with solubilizing tissue, our experience with human AD pathological tissue shows that STOMP of formalin-fixed archival tissue is possible. The STOMP technique is ideally suited for determination of the compositions of intracellular protein inclusions seen in ALS and FTD. While the principle

**Figure 6**. Microphotographs of Synaptophysin (**A**), VAMP2 (**B**) and SNAP25 (**C**) immunohistochemistry on the brain of human Alzheimer's disease cases. The amyloid plaques (arrow) are positive for VAMP2 and SNAP25, whereas synaptophysin does not stain the core of the plaques and punctate lobular profiles decorating the plaque are positive for synaptophysin. Scale bar is 30 µm.

components of these inclusions, such as TDP-43[9], FUS (*Kwiatkowski et al., 2009*; *Vance et al., 2009*), and dipeptide repeat proteins (*Mori et al., 2013*) are known, determination of the secondary components of the inclusions could shed light on disease etiology.

In developing the STOMP technique, we have focused on identification of photo-tagged proteins, however, the nonspecific nature of benzophenone photochemistry should cause photo-tagging of nonprotein components as well. In this regard, the STOMP technique can easily be modified to identify the RNA composition of biological features. RNA-containing subcellular structures such as P-bodies and stress granules have been implicated in neurodegenerative disease (*Li et al., 2013*); these structures are of suitable size for STOMP, and there is a realistic possibility of determining jointly their proteomic and transcriptomic composition with this technique. The application of this technique to many other aspects of cell biology research is also conceivable.

## Materials and methods

Unless otherwise specified, all dry reagents are from Sigma–Aldrich, St. Louis, MO. All organic solvents are from Caledon Laboratories, Georgetown, ON.

### Photo-tag synthesis

The 6HisBP peptide (4-benzoyl-benzamidyl-Gly-His-His-His-His-His-His-amide, *Figure 1*) was synthesized using FMOC-His(Trt)-OH, FMOC-Gly-OH (Avanced Chemtech, Louisville, KY) and 4-benzoylbenzoic acid; all synthesis steps were carried out in N,N-dimethylformamide (DMF). FMOC cleavage employed 2% 1,8-diazabicyclo[5.4.0]undec-7-ene (DBU); activation and coupling used 0.5 M N,N-diisopropylethylamine (DIPEA) and o-(7-azabenzotriazol-1-yl)-N,N,N′,N′-tetramethyluronium hexafluorophosphate (HATU, Applied Biosystems, Waltham, MA). The peptide was cleaved from the resin by incubating in 5 ml of a solution containing trifluoroacetic acid, thioanisole, ethanedithiol, and anisole in the volume ratio of 90:5:3:2, for 2 hr at room temperature, then precipitated by dropwise addition to 40 ml −70°C diethyl ether. The crude 6HisBP was washed five times with cold diethyl ether to remove residual cleavage reagents, dried under nitrogen, redissolved in ultrapure water, frozen, and lyophilized (Freezone 4.5, Labconco, Kansas City, MO) to remove any remaining volatiles.

The presence of 6HisBP was confirmed by ESI-MS.

### Murine tissue sectioning and preparation

Transgenic CRND8 (TgCRND8) mice express a transgene that encodes human amyloid precursor protein (APP695) harboring both the 'Swedish' double mutation KM670/671/NL and the additional 'Indiana' point mutation V717F; they develop dense-cored amyloid plaques and neuritic pathology by 5 months of age (*Chishti et al., 2001*). Animals were sacrificed at 8 months and their brains were

embedded in OCT medium for frozen sectioning onto glass slides. Sections were post-fixed by immersion for 10 min in −20°C methanol, air dried at room temperature, and stored at −70°C.

Brain sections were treated with DEPC to covalently modify endogenous histidyl residues and abolish their capacity to bind to nickel. Sections were rehydrated by immersion in DEPC buffer (10 mM NaHCO₃, 10 mM NaH₂PO₄, pH adjusted to 6.0 with hydrochloric acid). An initial working solution of 1 M DEPC in methanol was prepared immediately before use and diluted in DEPC buffer to 10 mM DEPC. Tissue sections were immersed in DEPC solution for 3 hr at room temperature, then washed twice with PBS.

Sections were stained with 1% ThS. Sections were soaked in two changes of 6HisBP working solution (12 mM 6HisBP, 10 mM sodium phosphate, pH 7.6), then coverslipped. For each STOMP sample, a 'dark' control was prepared from an adjacent section.

All experiments were performed in accordance with the guidelines imposed by the University of Toronto Animal Care Committee and Canadian Council for Animal Care.

## Microscopy and photoactivation

Microscopy employed Zeiss LSM 510 and LSM 710 confocal microscopes, equipped with two-photon infrared (Coherent Chameleon, tuned to 720 nm) laser sources. ThS fluorescence was excited at 488 nm and imaged confocally through 505 nm longpass or 500–530 nm bandpass filters. The nominal pixel spacing was 0.44 .44× 0.44 µm using a 20× 1.20 NA water immersion objective lens. Masks containing only amyloid plaques were created using the following four-step process in ImageJ (*Schneider et al., 2012*). High-frequency noise in the ThS fluorescence image was removed using ImageJ's 'Smooth' function, and vascular amyloid deposits—if any—were manually excluded from each fluorescence image. Dense-cored plaques were highlighted with the 'Threshold' tool. Finally, the 'Erode' method was used to trim a single-pixel (0.44 µm) margin from the mask edge, to ensure that only material definitely within the plaque would be photo-tagged. The resulting mask image was saved as a 'text image'.

Using a custom macro for controlling the two-photon microscope (*Pham et al., 2007*) each pixel in the mask image was irradiated six times, for 4 ms on each pass. A detailed description of the STOMP macro interface available in *Source code 1*. The total volume photo-activated was approximately $2 \times 10^6$ µm³ per sample ($1 \times 10^6$ µm³ per section).

## Solubilization and affinity purification

Each section was lifted from the slide using a clean, sterile razor blade and transferred into a 1.5-ml polypropylene microcentrifuge tube. For methanol-fixed murine tissue, 200 µl of solubilization buffer containing 2% sodium dodecyl sulfate (SDS, Bio-Rad, Hercules, CA), 8 M urea, 250 mM NaCl, 10 mM sodium phosphate buffer (pH 8.2), 10% (wt/vol) glycerol, and 2% β-ME was added to each tube. Tubes were immersed in a boiling water bath for 5 min, sonicated for 5 min in a Branson 1210 bath sonicator (Branson Ultrasonic Corporation, Danbury, CT), incubated at room temperature for 45 min, and returned to boiling water for an additional 5 min to fully solubilize the tissue sample.

For solubilization of formalin-fixed human tissue, 400 µl of a modified solubilization buffer containing 200 mM Tris (in addition to the ingredients noted above, as a formaldehyde scavenger) was used. Formalin-fixed sections were held at 90°C for 60 min. Each sample was transferred to a 15-ml polypropylene centrifuge tube and combined with a 20-fold excess of room-temperature dilution buffer (8 M urea, 250 mM NaCl, 10 mM sodium phosphate buffer, 1 mM imidazole) to dilute the SDS, β-ME, and Tris to concentrations compatible with Ni-NTA agarose bead binding. 80 µl of Ni-NTA agarose bead slurry (Qiagen, Valencia, CA) was added, and the mixture was tumbled overnight at room temperature to allow the His-tagged proteins to bind to the beads.

After binding, the beads were pelleted by centrifugation at 500×*g* for 30 s. The supernatant was discarded, and the beads were resuspended in 1 ml wash buffer (6 M urea, 0.1% SDS, 250 mM NaCl, 10 mM phosphate at pH 8.2, 0.1% β-ME, 1 mM imidazole) and tumbled for 2 min. This wash was repeated three times. During the final wash step, each sample was divided into two portions: one reserved for mass spectrometry, and one for gel electrophoresis and silver staining.

## Mass spectrometry

After recovery, the proteins captured on the surface of the beads for each sample were solubilized in 8 M urea with 50 mM Tris at pH 8, reduced by 5 mM DTT for 1 hr, and alkylated with 10 mM iodoacetamide for 45 min in darkness at room temperature. Proteins were digested at RT with Endoproteinase Lys-C (Roche) for 6 hr first, and then diluted ninefold into ammonium bicarbonate buffer before adding sequencing grade trypsin (Promega) overnight. After acidification to 1% formic acid (FA), mixtures were desalted using disposable Toptip C-18 columns (Glygen) and the eluted peptides lyophilized to dryness.

Peptides were loaded and separated on sequential reverse phase micro-capillary liquid trap and analytical columns using an EASY-nLC nanoflow pump system (Proxeon). The micro-capillary trap column was constructed in a 25 mm × 75 mm silica capillary packed with 5 µm Luna C18 stationary phase (Phenomenex). The analytical column was constructed in a 100 mm × 75 µm silica capillary, with a fine tip pulled with a column puller (Sutter Instruments), packed with 3 µm Luna C18 stationary phase. Separation was performed in 105 min using an organic gradient w consisting of buffer A (5% acetonitrile with 0.1% FA) to buffer B (95% acetonitrile with 0.1% FA) at a flow rate of 300 nl/min starting with a gradient of 2–6% buffer B in 1 min, followed by 6–24% in 74 min, 24–90% in 16 min, then 90% buffer B for 5 min and 90–0% in 1 min and finally 0% buffer B in 8 min. Eluted peptides were electrosprayed from a nanospray ion source (Proxeon) directly into a high-performance Orbitrap Velos hybrid tandem mass spectrometer (ThermoFisher Scientific). 10 data-dependent collision-induced dissociation (CID) ion trap scans (centroid mode) were automatically acquired simultaneously with one high resolution (60,000 FWHM resolution) full scan (profile mode) mass spectra. A dynamic exclusion list was enabled to exclude a maximum of 500 ion targets for 22.5 s.

RAW data files were extracted with the ReAdW program and submitted to database search using SEQUEST (v2.7) against UniProt/SwissProt protein sequence FASTA file containing 22,491 human proteins as well as an equivalent number of reversed decoy proteins (to estimate the empirical false discovery rate). Search parameters were set to allow for one missed cleavage site and one fixed modification (+57 on cysteine) using a precursor tolerance of 3 m/z. Matched peptides were further filtered at the precursor ion mass accuracy level using a 20 ppm cut-off, while protein hits were compiled and filtered using the StatQuest program with a minimum confidence threshold of 99%.

## Gel electrophoresis and silver staining

Silver staining employed an adaptation of the method of *Shevchenko et al. (1996)*. Gels were fixed in 6% formaldehyde and 25% ethanol for 1 hr and washed with water. Gels were sensitized for 45 min in 50 ppm sodium thiosulfate solution, soaked in 0.1% silver nitrate for 45 min, rinsed with ultrapure water, and developed for approximately 1 min in a solution of 2% sodium carbonate with 0.037% formaldehyde. Development was stopped with 50 mM EDTA. Silver-stained gels were photographed with an EOS 550D digital camera (Canon, Tokyo, Japan).

## Immunofluorescence and immunohistochemistry

TgCRND8 brain sections were stained with 1% ThS and blocked overnight with 10% fetal bovine serum (FCS) in PBS. Sections were incubated with primary antibodies against β-amyloid peptide, SNAP25, synapsin 1, synaptophysin, syntaxin 1, α-synuclein, V-type proton ATPase subunit B (ATP6V1B2) or VAMP2 at the dilutions indicated in *Supplementary file 4*, and then with the appropriate Texas Red-labeled secondary.

Formalin-fixed, paraffin-embedded sections of human brain from deceased AD patients were incubated with synaptophysin, SNAP25, and VAMP2 antibodies at the dilutions indicated in *Supplementary file 4*. After incubation with secondary antibodies, immunoreactivity was developed using diaminobenzidine and counterstained with hematoxylin.

Tissues from AD cases were collected with patient consent and handled under protocols approved by the University Health Network.

## Photo-tagging volume measurement

TgCRND8 brain sections (frozen sectioned, methanol post-fixed, DEPC treated) were soaked in 6HisBP solution, and a single pixel spot was photo-tagged using the STOMP macro as described above. Unbound photo-tag was then washed out of the section with several changes of PBS. Sections

were blocked with 5% bovine serum albumin, incubated with an anti-hexahistidine antibody, and fluorescently labeled with an Alexa 488-conjugated secondary antibody.

A high-resolution confocal z stack of immunofluorescence images was collected using a 63× 1.4 NA oil-immersion objective. Maximum intensity projections through the resulting z stacks were used to quantify the extent of the photo-tagged spot. Fluorescence intensity profiles through the center of the spot in x, y, and z directions were extracted using ImageJ and fitted with a Gaussian curve using OriginPro 8.5.0 (OriginLab, Northampton MA) to determine the width (FWHM) of the spot in each dimension.

## Acknowledgements

For helpful discussions, we thank Neil R Cashman (University of British Columbia), William S Trimble (University of Toronto), and Derek Van Der Kooy (University of Toronto). Miria Bartolini and Judy Cathcart at AOMF are thanked for microscopy support. Funding was from Bernice Ramsay Discovery Grants from the ALS Society of Canada, a David G. Dewar Research Grant from the Alzheimer Society of Canada, and a grant from the Canadian Consortium on Neurodegeneration in Aging (CCNA).

## Additional information

### Funding

| Funder | Grant reference | Author |
| --- | --- | --- |
| ALS Society of Canada | Bernice Ramsay Discovery Grant | Rishi Rakhit |
| Canadian Consortium on Neurodegeneration in Aging | | Kevin C Hadley, Yulong Sun |

The funders had no role in study design, data collection and interpretation, or the decision to submit the work for publication.

### Author contributions

KCH, RR, HG, Conception and design, Acquisition of data, Analysis and interpretation of data, Drafting or revising the article; YS, JENJ, Conception and design, Acquisition of data, Analysis and interpretation of data; JML, Contributed unpublished essential data or reagents; L-NH, Acquisition of data, Analysis and interpretation of data, Contributed unpublished essential data or reagents; AE, Conception and design, Analysis and interpretation of data, Drafting or revising the article; AC, Conception and design, Acquisition of data, Analysis and interpretation of data, Drafting or revising the article, Contributed unpublished essential data or reagents

### Author ORCIDs

Yulong Sun, http://orcid.org/0000-0002-6665-677X

### Ethics

Human subjects: The work presented was performed in compliance with recognized international standards, including the International Conference on Harmonization (ICH), the Council for International Organizations of Medical Sciences (CIOMS) and the principles of the Declaration of Helsinki. Use of human tissue was in accordance with the University Health Network Research Ethics Board. The Human brain samples were collected as part of the Canadian Brain Tissue Bank (CBTB). At the time of collection, informed consent was obtained.

Animal experimentation: This study was performed in strict accordance with the University of Toronto Animal Care Committee Guidelines.

## Additional files

### Supplementary files

• Supplementary file 1. All proteins identified by STOMP from TgCRND8 mouse plaques.

- Supplementary file 2. High-abundance proteins in non-plaque regions of the TgCRND8 mouse brain identified and retrieved by STOMP. Table is sorted in descending order of abundance, by normalized spectral counts.

- Supplementary file 3. 26 enriched proteins enriched in senile plaques relative to non-plaque regions detected by *Liao et al. (2004)* and *Söderberg et al. (2006)*.

- Supplementary file 4. Antibodies used for immunofluorescence and immunohistochemistry.

- Source code 1. Custom STOMP macro.

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
