## [Decision Letter]

Thank you for submitting your work entitled “Determining composition of micron-scale protein deposits in neurodegenerative disease by spatially targeted optical microproteomics”, for peer review at *eLife*. Your submission has been favorably evaluated by Huda Zoghbi (Senior Editor) and two reviewers, one of whom, Jeffery Kelly, is a member of our Board of Reviewing Editors.

We are pleased to inform you that your paper has been accepted for publication in *eLife* with minor revisions. In preparing your revised manuscript for publication please consider the minor comments of both reviewers, although our preference would be to leave the Introduction as it is.

*Reviewer #1 (General assessment and major comments):* This manuscript from Hadley and co-workers describes an exciting new technique called STOMP, which promises to revolutionise targeted microproteomics. The concept is very simple and at the same time very clever. Regions of interest in a biological specimen prepared for imaging are selected on the basis of the distribution of a suitable marker labelled with an antibody or fluorescent protein, and then a tag (in this case hexahistidine) is photo-crosslinked to the proteins distributed in this area using a multiphoton microscope. Upon solubilisation of the fixed sample on the slide, tagged proteins are purified using a nickel matrix and identified by mass spec. This is an ingenious alternative to labelling approaches such as BioID and Apex, with the important advantage that STOMP does not require expression of recombinant proteins in the cell of interest. In addition, the authors demonstrated that STOMP can be used in formalin-fixed tissue, a feature which great expand its application potential.

The authors apply this method to the analysis of amyloid plaques both in mouse tissue and in post-mortem specimens providing proof-of-principle of the reliability of the technique. Even though this paper did not strictly provide a novel mechanistic discovery, a method like STOMP can change the way we approach fundamental biological problems and therefore represents a major advance to the life science toolbox.

All my concerns are minor and deal more with the presentation of the data (see below). However, one possible major addition to this work is providing a proof-of-principle that this technique can be used with nucleic acids as well (e.g. RNA) as mentioned in the Conclusions.

*Reviewer #1 (Minor comments):* The Introduction is a little bit overlong, and should be prioritised. I suggest focussing on Abeta and the data already in the literature about the composition of amyloid plaques. Consideration about BioID and Apex may be better placed in the Discussion.

In the subsection “Validation of the STOMP results in TgCRND8 mice with immunofluorescence and immunohistochemistry”, the discussion on the impact of Abeta on synaptic function and synaptic protein distribution should be better organised.

Figure 1: consider making the blue colour more visible (panel A) and the chemical structure bigger and clearer.

Figure 4: The top panels are not very effective since they show virtually no staining. Consider including a different area in which staining for these proteins is present.

*Reviewer #1 (Additional data files and statistical comments):* The STOMP macro and the raw MS/MS data should be made available via a public server.

*Reviewer #2 (General assessment and major comments):* In Spatially targeted optical microproteomics (STOMP), the method under development in this manuscript, laser light from the microscope is used not only to image the fluorescently-stained specimen, but also to photochemically crosslink affinity purification ligands to protein components within pathological deposits (see Figure 1) to characterize the proteome within the imaged region. In principle a (1 micrometer) 3 volume can be imaged, but often a greater volume is needed to detect protein components with confidence. Nonetheless a much smaller volume is used herein that in laser dissection capture mass spectrometry-currently being used by the Mayo Clinic commercially to identify the amyloid composing the fibrils in systemic amyloidoises.

The photo-affinity ligand or photo-tag used here is the peptide 4-benzoylbenzyl-Gly-His_6_-CONH. A confocal image is used to generate a list of spatial coordinates-a “mask” file-identifying every point in the tissue containing pathological protein deposits. Two-photon microscopy is then used to photoexcite the benzophenone in a 1 micron3 (10,000 Å)3 volume as a minimal volume. Nickle affinity chromatography is then used to isolate the modified proteins. There are established endogenous proteins that bind to Nickle affinity chromatography resin supports, thus the permeabilized tissue sample is pretreated with diethylpyrocarbonate to destroy the binding of these proteins to the affinity resin via His modification before photoconjugation. The tagged proteins are dissolved in 2% SDS, 8M urea and β-mercaptoethanol, separated by gel electrophoresis and using a parallel sample identified by LC-MS/MS.

The authors used the TgCRND8 murine model featuring a mutated APP found in familial AD patients, employing thioflavine S to detect amyloid. In and around amyloid deposits from these transgenics and from formalin-fixed human brain tissue, it was found that v-ATPase, dynamin, SNAP25, VAMP2, synapsin, clathrin, ankyrin were enriched, and with the exception v-ATPase and dynamin, this study reveals they are enriched in or surround Abeta amyloid fibrils.

The authors demonstrate they can determine the proteome composition of micron-scale microscopic features in tissue samples of 500μmx500μm in an 8μm-thick tissue section, which is a significant advance. The proteomic data invoke Abeta aggregation and dysfunction of the synapse, which is supported by other methodology besides the STOMP method, which is the focus of this paper.

This is a very well thought, thorough study that in my mind demonstrates proof-of-principle, and for that reason is a strong contender for publication in *eLife*.

*Reviewer #2 (Minor comments):* The authors argue that the hexahistidine benzophenone tag is of similar size to Congo red and should have a similar ability to penetrate biological specimens which have been fixed and permeabilized using methanol. ClogP or other parameters along with a MW less than 750 typically predict cell permeability. In the future, why not use a small molecule high affinity binder to the nickel resin having more favorable physical chemical properties than hexa-His? This would allow this approach to be extended to intracellular amyloid in tissues.

Thioflavin S is not very specific for amyloid fibrils in tissue, which is a limitation the authors should mention in a revised manuscript.

---

## [Author Response]

Reviewer #1 (Minor comments): *The Introduction is a little bit overlong, and should be prioritised. I suggest focussing on Abeta and the data already in the literature about the composition of amyloid plaques. Consideration about BioID and Apex may be better placed in the Discussion.*

We agree with the editors that the Introduction, while somewhat long, sets the stage for our technological and scientific advances. We have left the Introduction as is, with the anticipation that readers with expertise in these areas may skip ahead to the Results and Discussion section of our paper.

In the subsection “Validation of the STOMP results in TgCRND8 mice with immunofluorescence and immunohistochemistry”, the discussion on the impact of Abeta on synaptic function and synaptic protein distribution should be better organised.

This section of the manuscript discusses possibilities for why we observe certain pre-synaptic proteins, but not others. Our two hypotheses are that: 1) some physical attribute of the proteins observed cause them to co-aggregate with amyloid plaques, or that 2) some physical property of synaptic vesicles as a whole cause them to bind to amyloid plaques. We changed the wording of the manuscript to better delineate these two possibilities.

Figure 1*: consider making the blue colour more visible (panel A) and the chemical structure bigger and clearer.*

It is unclear whether the reviewer here was looking at a full-resolution image here or a compressed image. In our hands, these images are clear; however, we believe this is an issue best handled at the copy-editing phase, and will make any changes requested during production.

Figure 4*: The top panels are not very effective since they show virtually no staining. Consider including a different area in which staining for these proteins is present.*

In our STOMP analysis of amyloid plaques, we found certain pre-synaptic proteins, but not others, even though many of these are commonly associated. In order to see whether STOMP merely missed these proteins, or whether these proteins were truly absent from the plaques, we carried out the analysis presented in Figure 4. In Figure 4, we show that certain pre-synaptic proteins not found by STOMP are in fact absent from the plaques using immunofluorescence with well-validated antibodies. We have added a line in the figure legend for Figure 4 to clarify this point.

Reviewer #1 (Additional data files and statistical comments): T*he STOMP macro and the raw MS/MS data should be made available via a public server.*

The STOMP macro will be made available as per request to the corresponding author, and we have made a note as such in the manuscript. The raw MS/MS data have been deposited to the ProteomeXchange public server under the accession # PXD002959, as now noted in the manuscript.

Reviewer #2 (Minor comments): T*he authors argue that the hexahistidine benzophenone tag is of similar size to Congo red and should have a similar ability to penetrate biological specimens which have been fixed and permeabilized using methanol. ClogP or other parameters along with a MW less than 750 typically predict cell permeability. In the future, why not use a small molecule high affinity binder to the nickel resin having more favorable physical chemical properties than hexa-His? This would allow this approach to be extended to intracellular amyloid in tissues.*

The reviewer refers to ClogP and MW<750 among Lipinski's ‘Rule of 5’ for predicting cell permeability. However, these rules are meant for living cells/tissues, rather than fixed cells; the STOMP technique specifically uses fixed cells. Our discussion about 6HisBP being similar size to Congo Red was simply to illustrate the fact that, unlike a large molecule such as an antibody, 6HisBP can freely diffuse into fixed tissues. In our experience, STOMP can be equally applied to intracellular targets without any modification to the procedures discussed in the manuscript.

Thioflavin S is not very specific for amyloid fibrils in tissue, which is a limitation the authors should mention in a revised manuscript.

It is true that all commonly used dyes to detect amyloid have some non-specific binding. Here we modified the manuscript (subsection “The STOMP technique”) to highlight these limitations and to briefly discuss the image processing steps that we undertook in order to overcome these limitations.